# Pump-underway ship intake: An unexploited opportunity for Marine Strategy Framework Directive (MSFD) microplastic monitoring needs on coastal and oceanic waters

Tania Montoto-Martínez[1¤]*, José Joaquín Hernández-Brito[1,2,3], Mª. Dolores Gelado-Caballero[1]

1 Environmental Technologies, Management and Biogeochemistry Research Group, University of Las Palmas de Gran Canaria, Canary Islands, Spain, 2 Atlantic International Research Centre, Azores, Portugal, 3 Oceanic Platform of the Canary Islands, Taliarte, Gran Canaria, Canary Islands, Spain

¤ Current address: Chemistry Department, Marine Sciences Faculty, University of Las Palmas de Gran Canaria, Gran Canaria, Spain

* tania.montoto@ulpgc.es

**Data Availability Statement:** Data are held in SEANOE repository: https://doi.org/10.17882/70158.

## Abstract

Broad scale sampling methods for microplastic monitoring in the open ocean waters remain a challenge in oceanography. A large number of samples is required to understand the distribution, abundance and fate of microplastic particles in the environment. Despite more than a decade of widespread study, there is currently no established time series of microplastic measurements and the research community is yet to establish a standardised set of methods that will allow data to be collected in a quick, affordable and interoperable way. We present a sampling technique involving the connection of a custom-built microplastic sampling device to the pump-underway ship intake system of a research vessel (RV) as an unexploited opportunity for oceanic monitoring needs concerning microplastic abundance and distribution. The method is cost effective, highly versatile and accurate, and is able to sample particles down to 50μm from opportunity platforms, thus contributing to an emerging area of study, and in particular helping to increase the monitoring reporting of data, and thereby serving as a valuable aid for the implementation of the Marine Strategy Framework Directive (MSFD). Sampling was performed during three consecutive oceanographic cruises in the subtropical NE Atlantic over a year, sampling subsurface waters (4 m depth) during navigation and while on coastal and oceanic stations. Microplastic particles were found in all stations and transects sampled. Fibres (64.42%) were predominant over fragments (35.58%), with the concentration values falling within the ranges of data reported for other areas of the Atlantic.

## Introduction

The ubiquity of microplastic pollution is backed up by a large number of studies and has been identified as an environmental threat of global concern [1–3]. The predominance of plastic

**Funding:** This work has been supported by a postgraduate research scholarship from the University of Las Palmas de Gran Canaria and carried out with funding from the European Regional Development Fund through the Madeira-Açores-Canarias Territorial Cooperation Operational Program (POMAC) 2014-2020 through the MARCET Project (MAC/1.1b/149).

**Competing interests:** The authors have declared that no competing interests exist.

amongst marine litter has also been clearly shown [4], representing up to 80% of marine litter composition according to assessments made in different marine regions [5,6].

Plastic pollution has been reported worldwide [7,8], in every environmental setting from the poles to the equator. The exponential growth of research into marine litter has resulted in many scientific articles describing its distribution, abundance and impacts [9]. Nowadays, microplastics have been explored and investigated on deep sediments [10], ice [11], rivers [12], marine biota [13,14], air [15], and in many other, often unexpected, locations [16–18]. Nonetheless, quantitative estimations are limited, and strong comparatives are limited due to the lack of standardization. With regards to geographical coverage, the southern hemisphere and remote areas are particularly understudied [19] and despite the oceans being where the majority of plastic was accumulates [7,20], relatively few studies focus specifically on the marine environment, owing to the inherent difficulties of oceanic sampling. Up to a 10% of the global plastic debris is estimated to enter the marine environment annually [21], but while coastal sediments have been widely explored, marine litter in oceanic waters is still under-investigated.

Numerous programs, committees, conventions and regulations worldwide have accompanied this declaration of plastic as an emerging pollutant [19]. At European level, the Marine Strategy Framework Directive (MSFD, 2008/56/EC), includes marine litter as a descriptor, and a Technical Subgroup on Marine Litter (TSG ML) was established to assess the Member States and support them on its implementation. Among the recommendations of this group, the need for data harmonization and the demand for further studies stand out as tasks to follow-up in order to pave the way for a better assessment of the microlitter abundance and distribution in oceanic waters. Many experts in the field agree that, although the impacts of this worsening problem might be challenging to study, there is sufficient evidence that increasing numbers of species are affected by marine litter with manifold consequences [22].

Due to its mobility, floating litter represents a pathway between different environmental compartments and plays an important role in the cycling of marine debris. It may be ingested by marine biota, act as a vector for the translocation of invasive species, or as precursors of microlitter through physical degradation, among other pathways [23]. Microlitter, defined as particles < 5 mm by the European Commission, is specifically considered in the Commission Decision (2017/848/EU), that lays down the criteria and methodological standards on good environmental status of marine waters and provides specifications and standardized methods for monitoring and assessment. This same document declares that surface layer on the water column should be further monitored.

The study of pelagic microplastics is a relatively new field (~5 years) [24], though their risk by ingestion has been demonstrated in different animal classes including marine mammals [25–27], seabirds [28,29], fishes [30–32] and zooplankton [33], posing an additional threat as a pathway to the transfer of organic chemicals to marine life [34]. However, the connectivity between research in different environmental compartments is rarely considered [24].

Sampling microplastic particles in water usually requires large sample volumes, and a range of trawls with variable net designs, mesh sizes and apertures are commonly used for studies in surface waters [35–37]. Net trawling has been recommended as a tool for microplastic research on the surface waters by the MSFD working group on Good Environmental Status (WG-GES) [38]. However, method validation and recovery checks have not been thoroughly applied to these methods, which may be resulting in erroneous estimations. Some disadvantages have been identified by the working group and other studies such as the difficulty of calculating the volume sampled due to water turbulence and their inefficiency in collecting smaller microplastics due to the mesh size limitation [39]. Furthermore, the window of operation is restricted to relatively calm sea conditions and a maximum tow speed of 3 knots [40].

Sampling surface waters is insufficient to understand the distribution and fate of these particles. It has been shown that turbulence due to wind [41] and other actions such as settling due to organic matter colonisation or other characteristics (including particle shape and size) may affect microplastic mobility within the water column [42]. There is some evidence that surface measurements underestimate microplastic abundance [41].

Sampling of microplastics from sub-surface waters can be achieved in a number of ways [36,43,44]. Some of these alternative methods are based on whole water sampling such as bulk water sampling, volume-reduced bucket or pump sampling, or niskin bottle sampling [45]. Though they have been less widely employed, the methods can be more widely applied at sea in rough conditions.

The pump-underway ship water intake has been used for this purpose in a number of published studies [46–50], and the development of methods that allow the sampling from ships of opportunity, such as onboard pumping and filtration systems, have also been encouraged by experts [23,51].

As the 6th International Marine Debris Conference Proceedings summarized, marine debris is a global problem that requires collaborative solutions on both the global and local scale [52]. Inspired by this, and bearing in mind the reports emitted by the TSG ML of the MSFD, the main aim of this study is to demonstrate the opportunities of a microplastic filtering device and a methodology based on the pump-underway system as a valid and useful method to increase microplastic monitoring and reported data in the short-term for oceanic waters.

In parallel, given the need to provide baseline data for the environmental status evaluation, and to put into practice our proposed approach, we provide the first baseline data of microplastic abundance and distribution in subsurface oceanic waters from a geostrategic area in the Macaronesian region (subtropical Northeastern Atlantic).

## Material and methods

### Sampling area

All the samples were collected during three oceanographic cruises to the European Station for Time Series in the Ocean Canary Islands (ESTOC; 29º10'N, 15º30'W) on 24–26 March 2018 (ESTOC 1803), 6–8 December 2018 (ESTOC 1812) and 21–25 February 2019 (ESTOC1902) commissioned by the Oceanic Platform of the Canary Islands (PLOCAN) on board the RV Ángeles Alvariño of the Spanish Institute of Oceanography (IEO). On each of the three cruises, two sampling modes are differentiated: (i) stationary mode (coastal stations, at the PLOCAN Test Site, and oceanic stations, at ESTOC) and (ii) navigation mode, as it is outlined in Fig 1 and Table 1. Both locations, the Test Site and the ESTOC station are part of the PLOCAN facilities for oceanic research [53].

### Sampling methods

Based on the gaps in knowledge and the needs for research development identified from a review of the current literature (see Table 2) and in recent reviews by Barrows [45], Mai et al. [36], Miller et al. [44] and Rocha-Santos and Duarte [43], a device was designed and built for opportunistic microplastic sampling with the following aims: (i) to help increase the number of observations and valid measurements to contribute to MSFD monitoring needs, in particular to Descriptor 10 (Marine litter); (ii) to enable an increase in knowledge concerning small-microplastics, through the retention of microplastic particles down to 50 μm; and (iii) to minimize contamination risk and ease operation on platforms of opportunity via a versatile, affordable and encapsulated design.

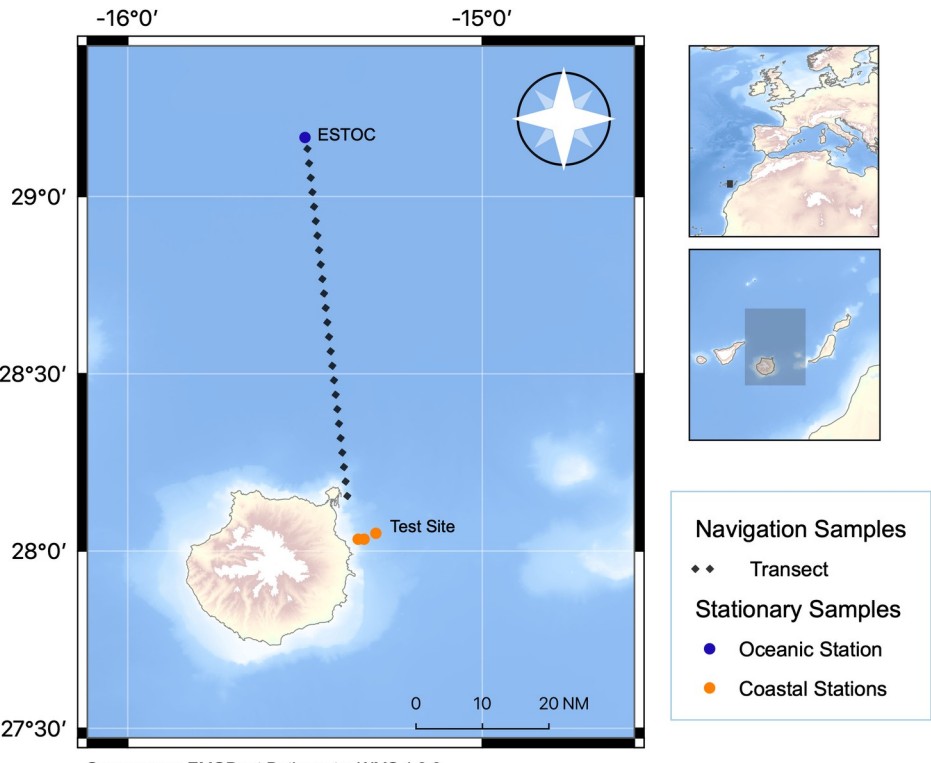

**Fig 1. Map of sampling locations repeated on the three research cruises.** (i) oceanic stations corresponding to the ESTOC station, (ii) coastal stations corresponding to three stations at the PLOCAN Test Site, 2 nautical miles off the coast of Gran Canaria, and (iii) navigation samples collected during the transit to and from ESTOC and the home port, a distance of approximately 60 nautical miles per transect. Coordinates are presented in Table 1. This figure was generated using the free and open source software QGIS [54].

In this way, the filtering device (Fig 2) consists of four stacked sieves (Ø = 100 mm) with mesh sizes of 300, 200, 100 and 50 μm. The design comprises a lid to prevent airborne contamination and a flowmeter, to ensure equal volume sampled in the different trials performed and to allow comparison of results among other studies. Several tests on board other research vessels and ships of opportunity, such as recreational vessels, were performed prior to the selection of this configuration for the device. In effect, the final design is a modification of the methods of Lusher et al. [50] and Enders et al. [48], where some changes were applied in order to increase the size distribution information provided and to be able to adapt the device to different sampling circumstances or conditions. Materials used were all standard parts that can be acquired in any hardware store at affordable prices, with the flowmeter being the trickiest to find. The device can be self-mounted by the researchers without additional skilled personnel

**Table 1. Coordinates of the sampling stations.**

| Oceanographic Campaigns | Sample type | Location | Latitude | Longitude |
|---|---|---|---|---|
| ESTOC 1803 ESTOC 1812 ESTOC 1902 | Stationary (Oceanic) | ESTOC | 29º 10' N | 15º 30' W |
| | Stationary (Coastal) | PLOCAN Test-Site (3 Stations) | 28º 3' N | 15º 18' W |
| | | | 28º 2' N | 15º 21' W |
| | | | 28º 2' N | 15º 20' W |
| | Navigation | Las Palmas Port to ESTOC (and back) | From 28º 7' N to 29º 10' N | From 15º 24' W to 15º 30' W |

**Table 2. Studies that performed similar continuous microplastic sampling pumping seawater on board research vessels.**

| Reference | Basin | Station Vs Transect | SW Intake supply | Depth (m) | Mesh size (s) (μm) | Sample vol. (L) | Total vol. (m³) | Microplastic concentration range (nº/m³) | Microplastic mean concentration (nº/m³) |
|---|---|---|---|---|---|---|---|---|---|
| Lusher et al. [52] | NE Atlantic | Transect [a] | Operating pump aboard | 3 | 250 | 2000 L | 940 | - | 2.46 ± 2.43 |
| Desforges et al. [49] | NE Pacific | Station and Transect | Pump-underway ship intake | 4.5 | 250; 125; 62.5 | [d] | - | 8.51–9180 | 2080 ± 2190 |
| Enders et al. [50] | NE Atlantic | Transect | Pump-underway ship intake | 0–3 | 300–10 [b] | 2600 ± 1300 | 60 | 13–501 | - |
| Setälä et al. [42] | Gulf of Finland | Station | Operating pump aboard | 0–0.5 | 300 and 100 [c] | 2000 | 2 | 0–1.25 and 0–6.8 [c] | 1.3 ± 1.9 and 0.6 ± 0.4 [c] |
| Cincinelli et al. [48] | Antarctica | Station | Pump-underway ship intake | 5 | - | 2000 | - | 0.0032–1.18 | 0.17 ± 0.34 |
| Kanhai et al. [51] | Atlantic Ocean | Transect [a] | Operating pump aboard | 11 | 250 | 2000 | 152 | 0–8.5 | 1.15 ± 1.45 |
| This work | NE Atlantic | Station and Transect | Pump-underway ship intake | 4 | 300; 200; 100; 50 | ~240 and ~2000 [e] | ~14 | 0–46.15 | 19.88 ± 15.92 |

[a] Does not differentiate when the RV stopped for regular station sampling.

[b] Smaller filter (10 μm) is switched to 50 μm due to clogging.

[c] The authors performed two separate trials with 300 and 100 mesh size filters respectively.

[d] Variable, depending on sampling time.

[e] Volumes for station and transect samples, respectively.

or specialized equipment needed on board (apart from the pump-underway system to collect the seawater).

**Stationary samples.** Using the microplastic filtering device described above, seawater samples were retrieved at the different sampling stations using the pump-underway ship-intake of the vessel, which takes the water at a constant flow rate from 4 meters depth, as it is shown in the set-up diagram in Fig 2. The overall sampling time at each station was around 35 ± 11 minutes–during which time a volume of 251 ± 20 L was sampled at an average flow rate of 7 L/min. The differences in the volumes sampled are corrected for by expressing the results in terms of number of particles per cubic meter. After acquisition of each sample, the filtering device was disconnected and each mesh was washed to a labelled screw top container with micro-filtered water (MilliQ, 1.2μm). Data from these measurements have been included with the other oceanic stations as part of the dataset.

**Navigation samples.** Sampling was also performed during two oceanographic cruises while on navigation from the home port (Las Palmas Port; 28º7'N, 15º24'W) to the oceanic ESTOC station and vice-versa. A total of 10472 L was sampled in the four separate transects, with a mean sampling volume of 2377 ± 895 L at an average flow rate of 5 L/min. The same procedure as on the stationary samples was followed after each transect was done, disconnecting the filtering device and washing each mesh to a labelled screw top container with micro-filtered water (MilliQ, 1.2μm).

## Contamination prevention and sampling procedural blanks

The filtering device and all filter meshes were thoroughly cleaned with MilliQ water before every use. Upon completion of sampling, the sieves were carefully withdrawn and the particles on the meshes carefully rinsed into their corresponding labelled container. A dress code of non-plastic material, cleaned work surfaces and washed forearms was applied to prevent contamination. To assess cleanliness, during sampling preparation and while rinsing and vacuum

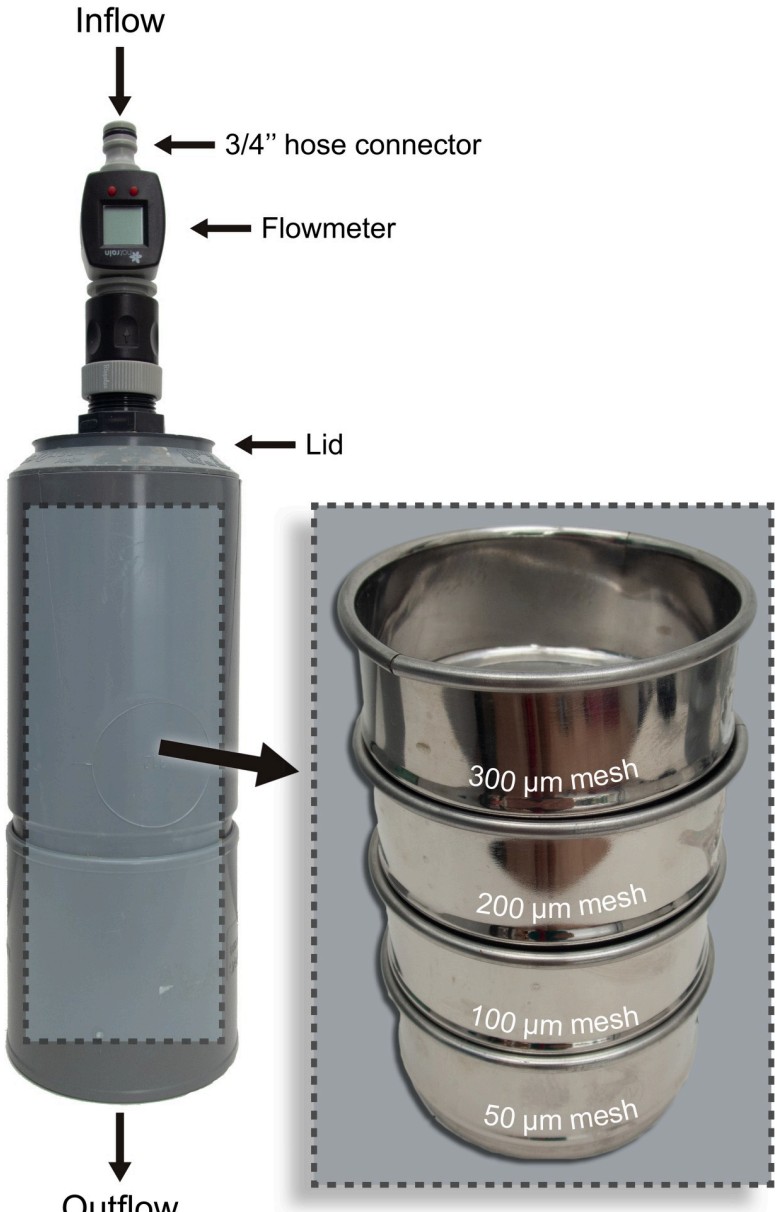

**Fig 2. Sampling device diagram.** The pump underway system of the RV Ángeles Alvariño had a tap connection at the wet lab to which the filtering device was connected with a ¾" hose. This figure shows real pictures of the sampling device used in this study, with the stacked sieves that it contains shown on the right.

filtering, open petri dishes with clean filter paper, were placed in direct proximity to the work area, providing a control of potential airborne contamination. These control filters were examined according to the same protocol as the target seawater filters.

Improvements to preventive measures, as recommended by Norén and Naustvol [55], have been shown to reduce airborne contamination. Common measures to prevent contamination during lab processing include (i) wearing cotton clothes while manipulating the samples, (ii) cleaning all containers using distilled water prior to its reuse, (iii) performing blank controls filtering MilliQ water, (iv) covering the sieves and other equipment to prevent air pollution

and (v) placing a clean petri dish with a filter paper next to the manipulation area to register possible airborne contamination [46,50].

## Laboratory analysis

Back on land, the water samples, one for every mesh size at each of the stations, was transferred to a glass beaker, rinsing thoroughly three times with MilliQ water. The glass beaker content was vacuum filtered using a 0.7 μm Whatman glass microfibre filter (GF/F, Ø = 47 mm). Each filter was placed on a petri dish and dried overnight before visual inspection, that was performed under a stereomicroscope (Nikon SMZ1000, 8-80X). The smallest particles and any particles whose origins were unclear were further observed under a metallographic microscope at up to 400X magnification (NIKON LV100POL), with any particles which were not clearly of plastic origin not taken into account for the analysis. Pictures and dimensions were recorded with a DS-Fi2 camera using NIS-Elements imaging software.

Each sample was processed under careful contamination prevention measures and kept covered when not being manipulated. Open petri dishes with clean filter papers were also placed next to the examination area, providing a second control of potential airborne contamination. This protocol is publicly available on dx.doi.org/10.17504/protocols.io.bcfyitpw.

Potential microplastics were identified following the guidelines of Lusher et al. [50], based on features such as their colour and form, and in the case of fibres, their thickness and three dimensional bending. Bearing in mind the potential bias in manually picking microplastics, only particles that were identified as plastic derivatives with high confidence were considered for the analysis. Confirmatory polymer analysis under a Raman microscope has been previously made for a parallel set of particles retrieved under similar conditions and showed a confirmation rate for microplastic identification of 95% (unpublished data).

The sorting data sheet comprised different categories such as films, pellets, foams, fibres and fragments; however, neither foams nor pellets or films were identified, and therefore the findings were categorised into two major groups (fibres and fragments). The results are given as number of particles per unit volume, to facilitate comparison between locations or times, and among different data users and providers [38]. Descriptive statistics were performed using R version 3.5.3 [56]. Means and standard deviations are reported throughout.

To verify repeatability, three consecutive samples were retrieved at the ESTOC oceanic station following the same procedure as the rest of the samples analysed on this study.

## Results

### Overview of findings

In this study, a total volume of 3525 L and 10472 L was sampled on the stations during three consecutive oceanographic campaigns, and during the four navigation transects, respectively. Microplastics were found in all samples taken at all stations and during all transects, identifying a total of 163 particles (5 mm– 50 μm): 105 fibres (64.42%) and 58 fragments (35.58%). Particle concentration ranged from 0 to 46.15 particles/m$^3$, with a mean value and standard deviation of 9.92±11.22 particles/m$^3$ (median: 5.5; 95% CI: 4.73–15.10), corresponding to 14.46±13.23 fibres/m$^3$ and 5.37±6.35 fragments/m$^3$ (Table 3).

Concerning the replicate experiment, 240 L were filtered with the pump-underway system in a consecutive way, obtaining a mean of 2.7 particles per replicate sample (SD: 1.2).

In many of the filters observed, fibres were found in agglomerations, as it is shown in Fig 3A. The picture in Fig 3B illustrates the typical appearance of the fragments identified, showing irregular edges and bright colour.

**Table 3. Microplastic particle concentrations expressed in number of particles/m³.**

| Mean concentration (Nº particles/m³) | Particle type | Navigations (n = 4) | Stations (n = 14) | Stations (n = 14) | | Total (n = 18) |
|---|---|---|---|---|---|---|
| | | | | Oceanic (n = 6) | Coastal (n = 8) | |
| | Fibres | 5.64 ± 5.35 | 16.98 ± 13.84 | 11.11 ± 8.08 | 21.39 ± 16.05 | 14.46 ± 13.23 |
| | Fragments | 3.97 ± 2.17 | 5.77 ± 7.13 | 7.06 ± 9.26 | 5.69 ± 6.29 | 5.37 ± 6.35 |
| | Total MP | 4.81 ± 3.88 | 11.38 ± 12.22 | 8.50 ± 8.49 | 13.54 ± 14.30 | 9.92 ± 11.22 |

Fibres are always present in greater numbers than fragments, both in stationary and navigation samples, as demonstrated in Fig 4, which shows the results separating by sample type: station (oceanic and coastal) and on navigation. Variability of abundance is in general also higher for fibres for all type of samples, with samples from coastal stations displaying the highest range (Table 3).

With regards to the procedural blanks, microplastics were not found in the air contamination controls set up on board during sampling manipulation. This suggests that no microplastics were introduced into the samples as a result of airborne contamination while operating the filtering device.

## Station vs navigation comparison

Of the 163 particles retrieved, 82 were obtained in samples while on navigation, and 81 while on the station. Despite the number of particles being almost equal, it must be noticed that the volume sampled was much larger for the navigation samples (10472 L) than sampled on station (3495 L). Considering only the fragments, the concentrations does not differ greatly between navigation and stationary samples (Table 3). In the case of fibres, they were approximately three times more abundant on the samples retrieved on station rather than during navigation ones, as is represented in Fig 4A.

With the microplastics collected while on station, a clear difference is apparent depending on the type of particle retained: fragments do not show significant differences between the two classes of station, with mean concentrations of 7.06±9.26 particles/m³ at oceanic stations compared with 5.69±6.29 particles/m³ at the coastal sites. In the case of fibres, there is a clear increase in fibres sampled at the coastal stations can be appreciated (Fig 4B).

With the microplastics collected during navigation, results indicate more similarity between fragment and fibre distributions in samples collected by this mode (Table 3).

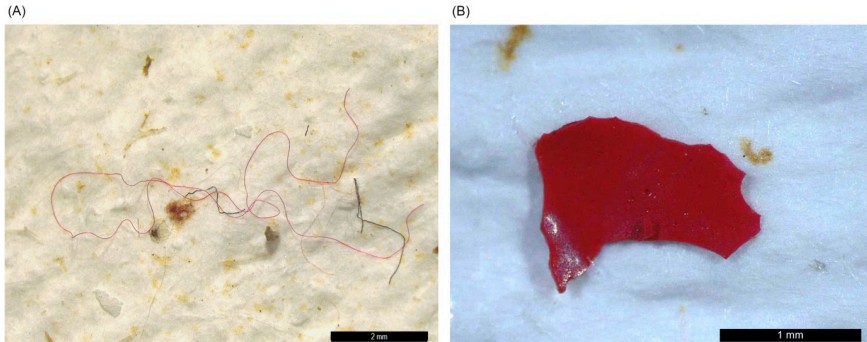

**Fig 3. Pictures illustrating the typical appearance of the microplastic particles found.** (A) An agglomeration of fibres, typically found on the samples; (B) A red microplastic fragment, showing characteristics for microplastic identification such as irregular edges and bright colour.

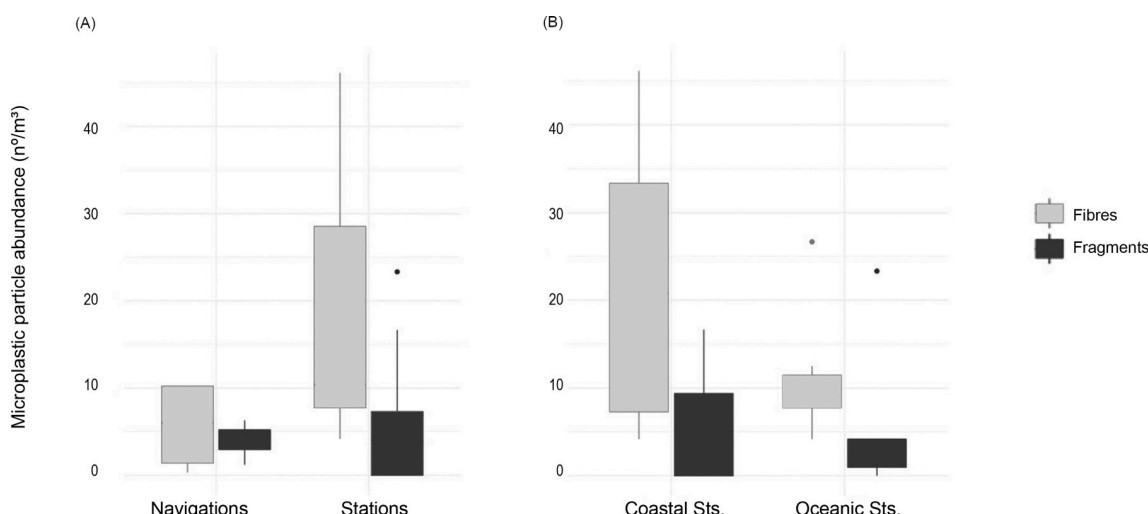

**Fig 4. Microplastic concentrations (nº/m³) depending on the sampling mode.** (A) On navigation vs on station; (B) Concerning station samples, coastal Vs oceanic stations.

### Colour and size distribution

Black, blue and red were the three predominant colours (>80%) of the total microplastic particles sampled, either on navigation or station, in coastal or oceanic waters (Fig 5). White or uncoloured particles were the next ones in abundance, followed by green and yellow in the case of the coastal and navigation samples.

Microplastic particles in the 1 to 3 mm size fraction were the most abundant ones retrieved on the samples taken on station. The fraction from 100 to 200 μm was the most abundant in the case of particles collected while on navigation sampling. Plastic particles larger than 5 mm were only retrieved on stations (Fig 6A), being all of them fibres (Fig 6B).

## Discussion

### Pump-underway as an opportunity to enhance monitoring

Towing a Neuston net is the most common sampling method for surface waters; however, the deployment of this infrastructure is not always an opportunity when at sea: it presents certain inaccuracy concerning the volume sampled and it is not suitable for the study of smaller microplastics due to the mesh size limitation. The underway water system of research vessels has been tested as an alternative method for microplastic sampling in a few studies to date, as it is shown in Table 2.

Subsurface seawater sampling and monitoring methods are often carried out by filling the bottles of a CTD Rosette sampler or similar bulk water sampling equipment. However, due to the higher volumes needed to study microplastic abundance, these methods are not the most suitable ones, and others, such as the manta trawl and the Neuston net can be used [40], having been widely employed in the last decade.

Ahead of this, the use of pumps as a low-cost system for circulating subsurface seawater through a deck tank while the ship is under way has been used for monitoring for many decades [57]. In fact, the pump-underway intake system, together with an in-line filtration system to prevent contamination, has been employed for this purpose in several studies up to date and proposed as a validated and effective technique that allows continuous sampling without interfering with the regular activity of the vessel on which it is operated [39,47–49]. This

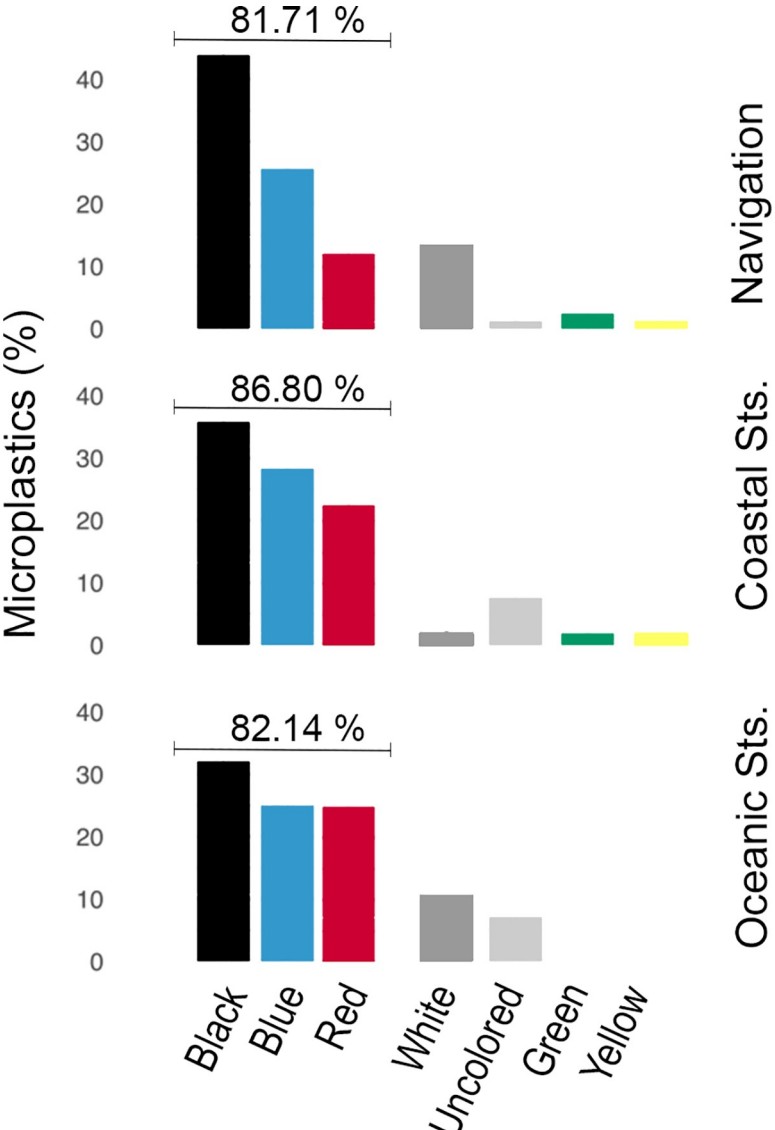

**Fig 5. Colour distribution for microplastic particles found.** Black, blue and red represent the predominant colours (>80%) in all three sample types.

method has the potential to be used for microplastic monitoring operations from a wider array of platforms of opportunity. Proof of its versatility is that it can even be used for surface micro-layer sampling from smaller boats [58].

In this study, we present a microplastic sampling device design and test a methodology based on the pump-underway ship intake system as an unexploited opportunity for oceanic monitoring needs. We consider that this approach has several practical advantages regarding its operation and wide spread use, such as:

i.  Cost-effective and affordable: The microplastic sampling device built and tested in this study was made with standard inexpensive parts that are easy to obtain. It can be self-mounted by the researchers without the requirement for specific tools or equipment to set it up. Though microplastic sampling nets are increasingly affordable, it must be noted that

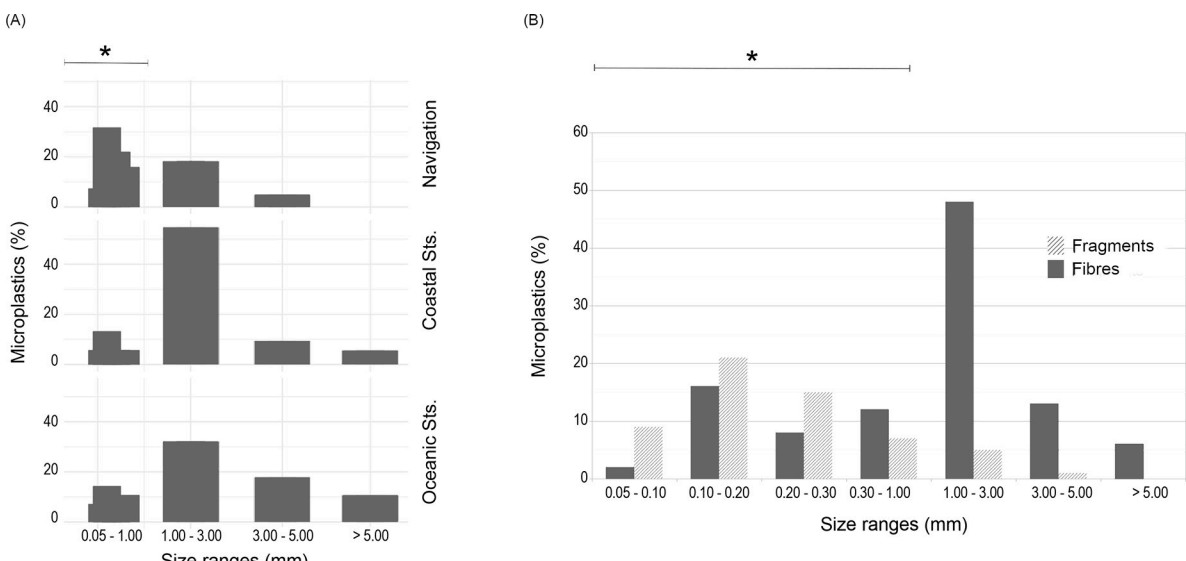

**Fig 6. Size distribution for microplastic particles found.** (A) Size distribution for microplastic particles found, separated by sample type; (B) Smallest size range separated into fractions from 0.05 to 0.10 mm; 0.10 to 0.20 mm; 0.20 to 0.30 mm and 0.30 to 1.00 mm, corresponding to the mesh sizes employed.

they also require other specialist equipment installed on the vessel and qualified personnel in order to be deployed.

ii. Opportunity: it can be employed to report data taking advantage of oceanographic cruises without interfering their regular vessel activities or limiting the speed, and therefore influencing in the cost of monitoring, where ship time is a considerable cost.

iii. Versatility: it can be adapted to different sampling requirements such as different depths or platforms, including recreational crafts. Additional filters can be mounted when passing through areas with more particulate matter in order to prevent clogging, and it can be scaled to the desired flow rate by altering the diameter.

iv. Accuracy: the method set up includes a flowmeter in the inlet of the device and therefore obtains information concerning microplastic abundance and distribution in a comparable way to other studies. At the moment, one of the main drawbacks of net trawling techniques is that the volume sampled is difficult to calculate, even in the designs provided with a flowmeter [59]. Consequently, errors are introduced when converting the amount of microplastic particles collected into concentrations per water volume [39].

v. Size range: it is suitable for small microplastic sampling -both fibres and fragments down to 50 μm- an emerging area of study due to the higher ecological risk that these smaller particles pose, and the most important for future investigations according to some authors [36,48] and reports [60,61]. In the case of nets, size fraction is often limited to 330 or 200 microns due to the water tension, fact that is thought to underestimate total MP concentrations by one to four orders of magnitude compared with samples that are filtered through much smaller mesh sizes (e.g. <100 μm) [62].

All these characteristics result in a considerable increase in the sampling window of opportunity, which combined with the versatility of possible platforms from which the device can be operated, lead to a real opportunity to enhance microplastic monitoring in oceanic waters. As a tool of interest for this purpose, it would serve to enhance the capability of sampling in the

under-explored marine environment, as encouraged in the report of the MSFD GES Technical Subgroup on Marine Litter entitled "Technical Recommendations for the Implementation of the MSFD Requirements" [23], where further research such as developing automated samplers to quantify microlitter in water is recommended.

## Microplastic filtering device evolution and performance

The design of the microplastic device used in this work has been progressively improved based on the results of several performance testing essays: different materials, mesh sizes and dimensions were tried until the most suitable combination was achieved. For instance, the original prototype, which was built with standard PVC rain gutters of 75 mm diameter and comprised a single mesh of 200 μm, was at times clogged, resulting in occasional leakages.

The filtering device, as depicted in Fig 2, allowed the study of the microplastic concentration on volumes up to 4000 L without clogging in transects of up to 60 nautical miles, which is the distance from the oceanic station ESTOC to the home port. In this way, the current design was suitable for continuous sampling the stations and also while on navigation, in both cases without interfering the ships' regular activity.

Additional advantages have been discovered in this work, such as the method being independent thanks to its tap connector, lid and flowmeter, allowing the researcher to perform other activities while the device is working without risk of affecting the sampling or introducing contamination.

## Comparison of reported data

As concluded by Monteiro et al. [24] in their paper on plastic pollution in islands of the Atlantic Ocean, there is a need for the scientific community to focus on the distant and far-separated oceanic islands, since these regions are important in terms of plastic pollution as a global environmental problem. The Canary Islands, located in the subtropical Northeastern Atlantic Ocean, 62 nautical miles west of Morocco at the closest point, are known for their high marine and terrestrial biodiversity. Microplastic studies in the area have been conducted in beaches [63,64] and, more recently, in fish [30]. However, with regard to seawater samples, this is the first study to report pelagic microplastics in the subsurface waters of the region.

Even though the aim of this study focuses on method validation, and despite the lack of standardization concerning microplastic monitoring in ocean samples being one of the main impediments to data comparability, we offer here some comparisons based on a careful selection of previous studies that were performed under similar conditions (Table 2).

Total microplastic concentration data for the subsurface ocean waters in the region (14.46 ±13.23 fibres/m$^3$ and 5.37±6.35 fragments/m$^3$) are higher than those reported for other areas of the Atlantic, including coastal areas. The lack of a method standardization with respect to sampling methods, size fraction classification, and other variables reduces the comparability of the results [65]. Nonetheless, our results are discussed below bearing in mind factors that might influence on the existing differences with other studies.

For instance, microplastic abundance along the north/south transect in the Atlantic Ocean performed by Kanhai et al. [49] ranged from 0 to 8.5 particles/m$^3$, showing the highest values offshore of Namibia (8.5 particles/m$^3$), and off the west coast of Morocco (6 to 6.5 particles/m$^3$). These amounts were sampled while on navigation, retrieving the seawater from 11 m depth with the pump-underway system. In the navigation samples in this study a mean value of 4.81±3.88 particles/m$^3$ was observed, which is within the range of the above-mentioned data. Differences, however, could be ascribed to the disparity on the sampling depth (11 vs 4 m) and the mesh sizes that were used (50 μm vs 250 μm).

Our results can be compared more easily to other research studies using the pump-underway system at similar sampling depths in different geographical areas. This is the case of the work performed by Lusher et al. [50] in the North East Atlantic and Cincinelli et al. [46] in the Antarctic, who obtained mean concentrations of 2.46±2.43 and 0.17±0.34 particles/m$^3$ respectively.

At a first glance, our results seem much higher than these reported values. However, considering only particles over 200 μm instead of the whole fraction, and bearing in mind only the samples taken while on navigation in order to perform a more accurate comparison, the concentration drops down to 2.03±2.09 particles/m$^3$.

The study carried out by Enders et al. [48] is the most comparable with this work, as the smallest mesh size employed was 10 μm, switching to 50 μm in regions with high suspended material content to prevent clogging. In their study, they also found microplastic in all stations sampled, ranging from 13 to 501 particles/m$^3$. Another of the conclusions drawn from this work was that microplastic increased steadily in numbers with decreasing size, this applied to both fibres and fragments. An observation that is also reflected in our study, in which the 55.21% of fibres are under 1mm, compared with the 64% of fibres under 900 μm measured by Enders et al. [48]. Because of the fragmentations of larger pieces of litter (legacy items), this fraction is expected to increase even if the introduction of larger items into the marine environment ceases [22]. The possibility to sample small microplastics is made possible by the pump-underway sampling method, which is less discriminative than other techniques such as net trawling, which cannot easily sample particles smaller than 300 μm.

Colour distribution is strongly consistent with the results of the previous studies listed in Table 2 that provided this information, with blue, black and red as the predominant colours. Blue and black are also among the predominant colours cited by several authors concerning fish ingestion articles [66–68], including the one published for the Canary Islands region by Herrera et al. [30]. This may also be related to the ease of visual identification: according to a study by Lenz et al. [69], blue is the most easily identified colour, followed by green and red.

Marine litter is generally more abundant in shallow coastal areas (<40 m depth) [70], with the exception of some accumulation zones in the open sea [71]. This is consistent with this study, where the concentration of microplastic particles sampled in the Test Site is significantly higher than the concentration obtained in oceanic samples: 13.54±14.30 particles/m$^3$, versus 8.50±8.49 particles/m$^3$. This difference was also noted by Cincinelli et al. [46] who sampled near-shore and off-shore the coastal area of the Ross Sea and linked the higher concentration near-shore to the influence of the scientific Mario Zuchelli Station and the effluent of the wastewater treatment plant. In our case, the Test Site is 2 nautical miles off the coast, just in front of the mouth of a ravine, and in the path of a marine current coming from the vicinity of the largest city on the island.

In the same way, relevant differences come up when comparing samples taking while on navigation (4.81±3.88 particles/m$^3$) as opposed to the samples taken on stationary mode (8.50±8.49 particles/m$^3$ for oceanic stations and 13.54±14.30 particles/m$^3$ for coastal stations). Again, if we consider separately by the type of particle, the difference does not seem significant for fragments. Further research is needed to confirm whether the hydrodynamics involved in the movement of the ship could influence the uptake of fibres by the pump-underway system of the vessel. This factor is not taken into account in the studies of Lusher et al. [50] and Kanhai et al. [49] which combined samples collected under a range of vessel speeds from 0 (stationary) to 10 knots (surveying). In this way, our results seem to indicate that there could be an underestimation of the fibres concentration while on navigation, and it could be that artefacts are produced by the sampling conditions during the collection of navigation samples.

## Conclusions

The use of the microplastic filtration device coupled to the pump-underway system shows great potential as a way of increasing microplastic pollution monitoring effort, retrieving information concerning microplastic abundance and distribution in a comparable way to previous studies, and providing, at the same time, further advantages: (i) the sampling device is fully made with standard materials, cost-effective and affordable, and it can be self-mounted by the researchers without additional skilled personnel or equipment needed onboard; (ii) it can be employed to report data taking advantage of oceanographic campaigns and other platforms at sea without interfering their regular activities; (iii) it is highly versatile; (iv) accurate and comparable; and (v) it is suitable to sample microplastic particles (both fibres and fragments) in a wider size spectrum, down to 50 μm.

In contrast to most published studies to date that report single records in time, this work represents the first attempt to microplastic monitoring on the area by repeating the same procedure over three consecutive oceanographic cruises in the same area. Microplastic particles were found in the total stations and transects sampled. Fibres (64.42%) were predominant over fragments (35.58%), with the concentration within the ranges of the data reported in other areas in the Atlantic, after calculations to correct for the lack of standardization among studies.

More research is needed in order to understand how navigation speed and sampling depth affect this sampling method, and further trials in the region to increase spatial and temporal coverage of measurements and to identify the possible influence of environmental factors on the concentrations and trends of microplastics in the Canary Islands region.

## Acknowledgments

The authors thank the crew of the RV Ángeles Alvariño, for their support in facilitating the research and Amelia Rodríguez-Rodríguez, head of the Archaeology Department at the ULPGC, for providing access to the microscope.

## Author Contributions

**Conceptualization:** José Joaquín Hernández-Brito, Mª. Dolores Gelado-Caballero.

**Funding acquisition:** Mª. Dolores Gelado-Caballero.

**Investigation:** Tania Montoto-Martínez.

**Methodology:** José Joaquín Hernández-Brito, Mª. Dolores Gelado-Caballero.

**Project administration:** Mª. Dolores Gelado-Caballero.

**Resources:** José Joaquín Hernández-Brito.

**Supervision:** José Joaquín Hernández-Brito, Mª. Dolores Gelado-Caballero.

**Writing – original draft:** Tania Montoto-Martínez.

**Writing – review & editing:** Tania Montoto-Martínez, José Joaquín Hernández-Brito, Mª. Dolores Gelado-Caballero.

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
