## [Decision Letter · Decision Letter 0]

6 Jan 2020

PONE-D-19-30825

Pump-underway ship intake: An unexploited opportunity for MSFD monitoring needs. First observations of microplastics on oceanic and coastal waters off the Canary Islands (Subtropical NE Atlantic)

PLOS ONE

Dear Mrs. Montoto-Martínez,

Thank you for submitting your manuscript to PLOS ONE. After careful consideration, we feel that it has merit but does not fully meet PLOS ONE’s publication criteria as it currently stands. Therefore, we invite you to submit a revised version of the manuscript that addresses the points raised during the review process.

I would particularly highlight the need to (i) make it clear that there is a potential bias in manual picking microplastics (ii) address the question of whether any confirmatory polymer analysis has been done or not (iii) provide a better explanation of the sampling design used and/or how its limitations may hamper result comparability, and (iv) add a description of the statistical analyses done as well as an indication of the estimate of variability used (SD, SE, other).

Should you wish to review the manuscript, ensure that the English in the manuscript is carefully revised to warrant it meets the requirements for publication in an international journal such as PLoS One.

We would appreciate receiving your revised manuscript by Feb 20 2020 11:59PM. To enhance the reproducibility of your results, we recommend that if applicable you deposit your laboratory protocols in protocols.io, where a protocol can be assigned its own identifier (DOI) such that it can be cited independently in the future. For instructions see: http://journals.plos.org/plosone/s/submission-guidelines#loc-laboratory-protocols

We look forward to receiving your revised manuscript.

Kind regards,

Gustavo M. Martins, PhD

Academic Editor

PLOS ONE

Journal Requirements:

5. We note that Figures 1 and 2 in your submission contain map images which may be copyrighted.

a. You may seek permission from the original copyright holder of Figures 1 and  to publish the content specifically under the CC BY 4.0 license. 

Reviewers' comments:

Reviewer's Responses to Questions

**Comments to the Author**

1. Is the manuscript technically sound, and do the data support the conclusions?

Reviewer #1: Partly

Reviewer #2: Yes

2. Has the statistical analysis been performed appropriately and rigorously? 

Reviewer #1: N/A

Reviewer #2: I Don't Know

3. Have the authors made all data underlying the findings in their manuscript fully available?

Reviewer #1: Yes

Reviewer #2: No

4. Is the manuscript presented in an intelligible fashion and written in standard English?

Reviewer #1: No

Reviewer #2: Yes

5. Review Comments to the Author

Reviewer #1: This study highlights the use of a ship’s underway pump system and size fractionated sieves to capture potential microplastics. Whilst this in itself is not new, as highlighted by the authors, the idea that it could be used for MSFD monitoring purposes is insightful. Comments regarding the manuscript are noted below;

Manually picking out suspected plastic particles from samples has many limitations, particularly in terms of bias. It is becoming more and more expected for environmental microplastics to be detected autonomously, for example via scanning or imaging mode using µFT-IR, especially for particles < 100 µm where the detection and analysis is highly skewed by manual selection (Lenz, R., Enders, K., Stedmon, C. A., Mackenzie, D. M., & Nielsen, T. G. (2015). A critical assessment of visual identification of marine microplastic using Raman spectroscopy for analysis improvement. Marine Pollution Bulletin, 100(1), 82-91.) This potential bias in picking should be made more transparent in this manuscript. In addition to this, there does not appear to have been any confirmatory polymer analysis (ie, via FT-IR or Raman) which is currently the minimum requirement for publication of environmental microplastic data. Without knowing whether an particle is actually synthetic or natural, microplastic concentrations cannot be reported and comparisons between other studies cannot be made.

Whilst the English in the manuscript is good, it is not quite up to the standard for publication in an international journal. I suggest that the authors enlist the help of someone who speaks English as their first language to edit the manuscript.

Some line specific comments;

Line 73 – I don’t understand why floating litter would play a role in the cycling of marine debris? Please calrify

Line 75 – micro-litter definition (particles < 5 mm) requires a reference as there are many different definitions. I would argue however, that a lower limit should also be specified so as not to include the realms of nano-particles.

Line 233 – Why was the number of foam particles observed not relevant? Please clarify.

Lines 239—240 - This should go into the Results section.

Reviewer #2: I carefully reviewed the manuscript “Pump-underway ship intake: An unexploited opportunity for MSFD monitoring needs. First observations of microplastics on oceanic and coastal waters off the Canary Islands (Subtropical NE Atlantic)” by Montoto-Martínez and colleagues. The paper presents an opportunistic sampling methodology for monitoring ocean microplastic using the pump-underway ship intake system of research vessels.

The introduction is appropriated, providing the reader a solid background on marine microplastic/litter research, while pointing out for its caveats and limitations. The objective is clearly stated. As mentioned by the authors, methods are not entirely new, and consist on a modification of similar sampling set ups by Lusher et al. (2014) Enders et al. (2015).

The method itself is clearly described (e.g. the pump system, filtering device apparatus and sample sorting) but the sampling scheme is somehow confusing preventing results to be properly analysed or discussed. There are different times of surveys, different “modes”, some samples come from stations, some come from replicates, other are transect derived. With such mix of variables, it is unclear how data is analysed. For instance, it seems that comparisons between the stationary and navigation approaches included samples from different campaign dates. In this case, any differences detected may have been due to temporal variability and not related to the approach itself. Data analyses need more clarification. Also, given the reduced number of samples, and/or the spatial variability nature of oceanic microplastics, means in Table 3 do not provide that much information e.g. almost all deviations are higher than means. With such a dispersion of data, there is likely any significant differences between sampling approaches and/or particle types. As I understand, and given the lack of a robust sampling scheme, the authors should put more emphasis on what the paper is really about: the usefulness of an opportunistic method to monitor ocean microplastic (which in fact is done in the first section of the discussion).

Other minor comments:

Title: I would avoid using abbreviations in the manuscript title

Ln43: avoid keywords already referred in the manuscript title

Ln55: sentence “...long list of even unsuspected places.” needs some references

Ln85: remove paragraph.

Ln98: replace WG by working group.

Ln105/106: there is no previous reference in text to MP abbreviation. Same in Table 2. Ln164: there is no need for the use of hyphens.

Ln175: If possible, I would complement (or replace) Fig. 2 with real pictures of the research vessel, pump system and filtering device apparatus.

Ln242-261: I would move this paragraph to discussion.

Ln:268: Identify which measure of dispersion is being used. In means, there is no indication whether the second figure is the standard deviation or the standard error (or indeed something else).

Ln291-305: Again, this comparison is obsolete if data is originated from distinct campaign times.

Ln375-445: The authors are the first to recognize that methods used for oceanic microplastic detection are diverse and that “no harmonization has been yet fully accomplished”. Moreover, they acknowledge that “The lack of a method standardization concerning sampling method, size fraction, and other variables reduce the comparability of

the results”. Yet, in this section they compare results with other studies, that use different methods, distinct approaches with no standardization between them whatsoever. I find this section very speculative and suggest its removal.

Overall, I find the study interesting and, provided that some minor changes are made, it should be considered for publication.

6. PLOS authors have the option to publish the peer review history of their article (what does this mean?). If published, this will include your full peer review and any attached files.

Reviewer #1: No

Reviewer #2: No

---

## [Author Response · Author response to Decision Letter 0]

5 Mar 2020

Dear Dr. Martins, 

Thank you for your letter and the opportunity to revise our paper. We appreciate your insightful comments and the invitation to submit a revised and improved version of the manuscript. For that, the suggestions offered by the reviewers have also been immensely helpful. 

In response to your comments, we have now (i) made clear that there is a potential bias in manual picking microplastics (Line 222) and (ii) have addressed the question of the confirmatory polymer analysis (Line 224), further discussed in the comments to the reviewers below. Figures 1 and 2 were improved in order to (iii) provide a better explanation of the sampling design used, and a new paragraph starting in Line 386 remarks on the limitations that may be hampering the comparability of data in the manuscript. Furthermore, (iv) a description of the statistical analyses and the indications of the used estimates were included (Line 231). 

We have also extensively revised the article to improve the style and grammar so it meets the requirements for publication in an international journal such as PLOS One. 

I would like to thank you for the recommendation concerning the deposit of the laboratory protocol, which is now available at protocols.io: dx.doi.org/10.17504/protocols.io.bcfyitpw

As required, specific journal requirements were revised, and the changes made are explained below following each of the individual comments: 

1. Compliance with PLOS ONE’s style requirements:

Thank you for this direction. There were indeed some changes that had to be performed in order to the manuscript to fully comply with the PLOS ONE’s style requirements. 

We have now carefully followed the style templates and made some minor but necessary changes such as adapting the title format, removing the keywords and ensure that the file name is also labelled as indicated for the submission of the revised manuscript. 

2. Language usage, spelling and grammar copyediting:

As suggested, we have had the manuscript reviewed by a native English speaker to warrant it meets the requirements for publication in an international journal such as PLOS One. Dr. Matthew Patey, a former co-worker in our research group, current marine researcher at the National Oceanographic Centre (Southampton) has edited the manuscript.

3. Permits obtained for the work: 

No specific permits were required to perform the work. The Oceanic Platform of the Canary Islands (PLOCAN), which is one of the institutions participating in this research, offers a test site area that covers a public domain marine area located off the Northeast coast of Gran Canaria with 23 km2 and maximum depths of 600 meters, a land headquarters and harbor facilities (https://www.plocan.eu/en/test-site/). 

Nonetheless, in order to make this point clear, the following sentence was included in the manuscript: (Line 124) - Both locations, the Test-Site and the ESTOC Station are part of the PLOCAN facilities for oceanic research. 

4. Data availability. 

Data was uploaded to SEANOE (SEA scieNtific Open data Edition) repository. The url was not ready at the time we sent the manuscript. Data is now available at: https://doi.org/10.17882/70158

5. Copyright issues on Figures 1 and 2. 

We would like to thank you for the extensive information provided concerning the options we had to address this Copyright issue. 

As there was also a comment from one of the reviewers encouraging the replacement of Figure 2 (or complementing it with real pictures), we have taken this opportunity to replace both Figure 1 and 2 for better explanatory ones. The new Figures comply with the Creative Commons Attribution License (CC BY 4.0) and other copyright guidelines that apply to articles published in PLOS One. Both figure captions have been updated with source information. 

I have included the reviewers comments immediately after this letter and responded to them individually, indicating exactly how we addressed each concern or problem and describing the changes made in the document. 

All three authors have approved the revised manuscript, which we have also included as a marked-up copy highlighting the changes made to the original version. 

We hope the revised manuscript will better suit PLOS One but are happy to consider further revisions, and we thank you for your interest in our research.

Sincerely, 

Tania Montoto

PhD Student on Marine Sciences

University of Las Palmas de Gran Canaria

 

Reviewer 1

General Comments

This study highlights the use of a ship’s underway pump system and size fractionated sieves to capture potential microplastics. Whilst this in itself is not new, as highlighted by the authors, the idea that it could be used for MSFD monitoring purposes is insightful. Comments regarding the manuscript are noted below. 

Response to reviewer: Thank you! We found your comments extremely helpful and have revised accordingly.

Manually picking out suspected plastic particles from samples has many limitations, particularly in terms of bias. It is becoming more and more expected for environmental microplastics to be detected autonomously, for example via scanning or imaging mode using µFT-IR, especially for particles < 100 µm where the detection and analysis is highly skewed by manual selection (Lenz, R., Enders, K., Stedmon, C. A., Mackenzie, D. M., & Nielsen, T. G. (2015). A critical assessment of visual identification of marine microplastic using Raman spectroscopy for analysis improvement. Marine Pollution Bulletin, 100(1), 82-91.) This potential bias in picking should be made more transparent in this manuscript. 

Response to reviewer: Thank you for raising this point. Indeed, there is a potential bias that we have now made clearer in the manuscript. 

Changes: Line 222 – Bearing in mind the potential bias in manually picking microplastics, only particles that were identified as plastic derivatives with high confidence were considered for the analysis.

In addition to this, there does not appear to have been any confirmatory polymer analysis (ie, via FT-IR or Raman) which is currently the minimum requirement for publication of environmental microplastic data. Without knowing whether a particle is actually synthetic or natural, microplastic concentrations cannot be reported and comparisons between other studies cannot be made. 

Response to reviewer: Confirmatory polymer analysis was conducted on the particles identified on this study. However, we have previously worked with micro FT-IR and micro Raman, doublechecking the correct identification of particles from previous oceanographic campaigns. Our results returned high rates (95%) of plastic particle confirmation, which are below the misidentification rate provided in previous studies (Lenz et al, 2015). 

Changes: 

Subtle changes to emphasize on Line 211: The smallest particles and any particles whose origins were unclear were further observed under a metallographic microscope at up to 400X magnification (NIKON LV100POL), with any particles which were not clearly of plastic origin not taken into account for the analysis

Line 224 – Confirmatory polymer analysis under a Raman microscope has been previously made for a parallel set of particles retrieved under similar conditions and showed a confirmation rate for microplastic identification of 95% (unpublished data).

Whilst the English in the manuscript is good, it is not quite up to the standard for publication in an international journal. I suggest that the authors enlist the help of someone who speaks English as their first language to edit the manuscript.

Response to reviewer: Dr. Matthew Patey, a former co-worker in our research group, native English speaker and marine researcher working at the National Oceanographic Centre (Southampton) has reviewed and revised this manuscript for style and grammar and to ensure it meets the requirements for publication in an international journal such as PLOS One.

Changes: The whole document was revised.

Specific Comments

Line 73 – I don’t understand why floating litter would play a role in the cycling of marine debris? Please, clarify.

Response to reviewer: The paragraph was reformulated, adding some specific examples in order to clarify the reasons. 

Although we have not included it in the manuscript, looking for more references on this issue, we came across a recently published study on the role of the plastisphere recycling the greenhouse gases (Cornejo-D’Ottone et al., 2020). 

Changes: Line 67 – Due to its mobility, floating litter represents a pathway between different environmental compartments and plays an important role in the cycling of marine debris. It may be ingested by marine biota, act as a vector for the translocation of invasive species, or as precursors of microlitter through physical degradation, among other pathways [23]. 

Line 75 – micro-litter definition (particles < 5 mm) requires a reference as there are many different definitions. I would argue however, that a lower limit should also be specified so as not to include the realms of nano-particles

Response to reviewer: This definition is from the European Commission. We have rephrased the sentence in order to make this clear. We agree that there should be a lower limit. Currently there is no established lower limit, but we hope that, in future documents and Decisions, a lower limit will be specified. 

Changes: Line 70- Microlitter, defined as particles < 5 mm by the European Commission, is specifically considered in the Commission Decision (2017/848/EU), […]

Line 233 – Why was the number of foam particles observed not relevant? Please clarify.

Response to reviewer: What we wanted to say is that although we used a classification system with five entries to discern among different potential particles (i.e. one for each of these types of particles: fragments, fibres, foams, films and pellets), in the end, we could not confirm any foam, film or pellet within the samples, so the final categories in this study were just two: fibres and fragments. This issue was now clarified in the manuscript.

Changes: Line 227 – The sorting data sheet comprised different categories such as films, pellets, foams, fibres and fragments; however, neither foams nor pellets or films were identified, and therefore the findings were categorised into two major groups (fibres and fragments).

Lines 239—240 - This should go into the Results section

Response to reviewer: We have moved the lines to the Results section, as suggested. 

Changes: Line 244 – Concerning the replicate experiment, 240 L were filtered with the pump-underway system in a consecutive way, obtaining a mean of 2.7 microplastic particles per replicate (SD: 1.2). 

Reviewer 2

General Comments

I carefully reviewed the manuscript “Pump-underway ship intake: An unexploited opportunity for MSFD monitoring needs. First observations of microplastics on oceanic and coastal waters off the Canary Islands (Subtropical NE Atlantic)” by Montoto-Martínez and colleagues. The paper presents an opportunistic sampling methodology for monitoring ocean microplastic using the pump-underway ship intake system of research vessels.

The introduction is appropriated, providing the reader a solid background on marine microplastic/litter research, while pointing out for its caveats and limitations. The objective is clearly stated. As mentioned by the authors, methods are not entirely new, and consist on a modification of similar sampling set ups by Lusher et al. (2014) Enders et al. (2015).

Response to reviewer: Thank you for your helpful comments and for taking the time to suggest improvements to our manuscript. 

The method itself is clearly described (e.g. the pump system, filtering device apparatus and sample sorting) but the sampling scheme is somehow confusing preventing results to be properly analysed or discussed. There are different times of surveys, different “modes”, some samples come from stations, some come from replicates, other are transect derived. With such mix of variables, it is unclear how data is analysed. 

Response to reviewer: We have improved Figure 1 in a way we believe it makes the sampling scheme much clearer.

Changes: 

- Resubmission of Figure 1. 

- We have also reformulated the paragraph where the sampling modes are explained: Line 121 - On each of the three cruises, two sampling modes are differentiated: (i) stationary mode (coastal stations, at the PLOCAN Test Site, and oceanic stations, at ESTOC) and (ii) navigation mode, as it is outlined in Fig 1 and Table 1. Both locations, the Test Site and the ESTOC station are part of the PLOCAN facilities for oceanic research [53].

For instance, it seems that comparisons between the stationary and navigation approaches included samples from different campaign dates. In this case, any differences detected may have been due to temporal variability and not related to the approach itself. Data analyses need more clarification. Also, given the reduced number of samples, and/or the spatial variability nature of oceanic microplastics, means in Table 3 do not provide that much information e.g. almost all deviations are higher than means. With such a dispersion of data, there is likely any significant differences between sampling approaches and/or particle types. As I understand, and given the lack of a robust sampling scheme, the authors should put more emphasis on what the paper is really about: the usefulness of an opportunistic method to monitor ocean microplastic (which in fact is done in the first section of the discussion).

Response to reviewer: Just as you say, comparisons include samples from different cruise dates. However, we do not establish any temporal or spatial analysis. As you well comment, this study does not focus on the distributions or concentrations of microplastic particles in the Canary Islands, but on the method itself. Our aim with this comparisons is to prove that the sampling device employed is useful for its operation while the research vessel is both on navigation or on dynamic positioning, and also when concentration ranges are either those of oceanic waters (less amount of particles) or coastal waters (generally more polluted). 

Other minor comments

Title: I would avoid using abbreviations in the manuscript title. 

Response to reviewer: The manuscript title has been reformulated and shortened. 

Changes: Line 1 - Pump-underway ship intake: An unexploited opportunity for Marine Strategy Framework Directive (MSFD) microplastic monitoring needs on coastal and oceanic waters. 

Ln43: avoid keywords already referred in the manuscript title. 

Response to reviewer: Thank you for bringing this up to our attention. We have realised that the journal formatting guidelines specify that keywords should be uploaded separately from the manuscript, so the revised manuscript no longer includes keywords. 

Changes: Line N/A - Keywords were removed from the revised manuscript document due to formatting requirements. 

Ln55: sentence “...long list of even unsuspected places.” needs some references. 

Response to reviewer: We have included several references that, in our opinion, illustrate the ubiquity of microplastics in the environment.

Changes: Line 49 – References numbered 16, 17 and 18 were added. 

Ln85: remove paragraph.

Response to reviewer: Done. 

Ln98: replace WG by working group.

Response to reviewer: Done

Changes: Lines 84 and 87.

Ln105/106: there is no previous reference in text to MP abbreviation. Same in Table 2.

Response to reviewer: MP abbreviation was replaced by the whole word in the three cases. 

Changes: Lines 94 and 95. And also in Table 2 (Line 148).

Ln164: there is no need for the use of hyphens. 

Response to reviewer: Hyphens were removed, thank you.

Changes: Line 155

Ln175: If possible, I would complement (or replace) Fig. 2 with real pictures of the research vessel, pump system and filtering device apparatus.

Response to reviewer: We understand and appreciate your suggestion. Changes on this figure were also required by the editor. 

Changes: Figure 2 was replaced for a new one including real photos. Consequently, the caption was also slightly changed: Line 167 - Fig 2. Sampling device diagram. The pump underway system of the RV Ángeles Alvariño had a tap connection at the wet lab to which the filtering device was connected with a ¾’’ hose. This figure shows real pictures of the sampling device used in this study, with the stacked sieves that it contains shown on the right

Ln242-261: I would move this paragraph to discussion.

Response to reviewer: Done.

Changes: The paragraph has been moved to the Discussion section and consequently the title introducing the microplastic concentration data report under the Results section was deleted (Line 235).

Ln:268: Identify which measure of dispersion is being used. In means, there is no indication whether the second figure is the standard deviation or the standard error (or indeed something else).

Response to reviewer: We have included more information on the statistical analysis.

Changes: Line 240 - Particle concentration ranged from 0 to 46.15 particles/m3, with a mean value and standard deviation of 9.92±11.22 particles/m3 (median: 5.5; 95% CI: 4.73-15.10), corresponding to 14.46±13.23 fibres/m3 and 5.37±6.35 fragments/m3 (Table 3).

Ln291-305: Again, this comparison is obsolete if data is originated from distinct campaign times.

Response to reviewer: We are aware that variability concerning ocean microplastics among different campaign dates is inherent in the nature of marine studies. As mentioned in the response to the previous comment, the aim of this section is to prove that the sampling device is useful for its operation no matter if the vessel is on navigation or on dynamic positioning, and on oceanic or coastal waters. 

Data presented is therefore referred to the performance validation of the sampling device under different environmental conditions, including hydrodynamic conditions. 

Ln375-445: The authors are the first to recognize that methods used for oceanic microplastic detection are diverse and that “no harmonization has been yet fully accomplished”. Moreover, they acknowledge that “The lack of a method standardization concerning sampling method, size fraction, and other variables reduce the comparability of “the results”. Yet, in this section they compare results with other studies, that use different methods, distinct approaches with no standardization between them whatsoever. I find this section very speculative and suggest its removal.

Response to reviewer: The comparisons we offer in the last section of the discussion have been carefully picked up, taking into account only selected studies that used similar sampling approaches (ie the studies included in Table 2). In such a way, we consider that the reported comparisons are an adequate strategy for the sampling device performance validation. 

Changes: We have added a clarifying paragraph to introduce this part of the discussion (Line 386): Even though the aim of this study focuses on method validation, and despite the lack of standardization concerning microplastic monitoring in ocean samples being one of the main impediments to data comparability, we offer here some comparisons based on a careful selection of previous studies that were performed under similar conditions (Table 2). 

Overall, I find the study interesting and, provided that some minor changes are made, it should be considered for publication

Response to reviewer: Thank you!

---

## [Editor Report · Decision Letter 1]

22 Apr 2020

Pump-underway ship intake: An unexploited opportunity for Marine Strategy Framework Directive (MSFD) monitoring needs on coastal and oceanic waters.

PONE-D-19-30825R1

Dear Dr. Montoto-Martínez,

We are pleased to inform you that your manuscript has been judged scientifically suitable for publication and will be formally accepted for publication once it complies with all outstanding technical requirements.

With kind regards,

Gustavo M. Martins, PhD

Academic Editor

PLOS ONE

Additional Editor Comments (optional):

I am pleased with the thorough revisions made by the authors and I find that the manuscript in now deemed suitable for publication in PLOS ONE as is.
---

## [Editor Report · Acceptance letter]

30 Apr 2020

PONE-D-19-30825R1 

Pump-underway ship intake: An unexploited opportunity for Marine Strategy Framework Directive (MSFD) microplastic monitoring needs on coastal and oceanic waters. 

Dear Dr. Montoto-Martínez:

I am pleased to inform you that your manuscript has been deemed suitable for publication in PLOS ONE. Congratulations! Your manuscript is now with our production department. 

With kind regards,

on behalf of

Dr. Gustavo M. Martins 

Academic Editor

PLOS ONE